# Conjunctivitis, the key clinical characteristic of adult rubella in Japan during two large outbreaks, 2012–2013 and 2018–2019

Hidetoshi Nomoto[1,2], Masahiro Ishikane [1,3]*, Takato Nakamoto[1], Masayuki Ohta[1], Shinichiro Morioka[1], Kei Yamamoto[1], Satoshi Kutsuna[1], Shunsuke Tezuka[4], Junwa Kunimatsu[5], Norio Ohmagari[1,2,3]

1 Disease Control and Prevention Center, National Center for Global Health and Medicine, Shinjuku-ku, Tokyo, Japan, 2 Collaborative Chairs Emerging and Reemerging Infectious Diseases, National Center for Global Health and Medicine, Graduate School of Medicine, Tohoku University, Sendai city, Miyagi, Japan, 3 AMR Clinical Reference Center, National Center for Global Health and Medicine, Shinjuku-ku, Tokyo, Japan, 4 Laboratory testing department, National Center for Global Health and Medicine, Shinjuku-ku, Tokyo, Japan, 5 Department of General Internal Medicine, National Center for Global Health and Medicine, Shinjuku-ku, Tokyo, Japan

* ishikanemasahiro@gmail.com

**Data Availability Statement:** All relevant data are within the manuscript.

**Funding:** The authors received no specific funding for this work.

## Abstract

### Background

Rubella virus infection mainly causes illness with mild fever, rash, and lymphadenopathy in children; however, the clinical characteristics of adult rubella are not well-known.

### Methods

An observational study was conducted to compare the characteristics between adult rubella and adult non-rubella among participants aged ≥18 years, with suspected symptomatic rubella. Participants were screened for rubella-specific IgM expression using an enzyme immune assay kit, at a tertiary care hospital in Japan during two outbreaks (January 2012–December 2013 and January 2018–March 2019). Adult rubella diagnosis followed strong positive or paired rubella-specific IgM expression or positive rubella-specific reverse-transcription-polymerase chain reaction. Patients aged <18 years or with clinically suspected rubella with weak or negative IgM expression were excluded.

### Results

Overall, 82 adult rubella and 139 adult non-rubella, with a median age (interquartile range) of 31 (25–41) years and 34 (27–42) years, respectively, were included. Multivariate analysis showed that conjunctivitis (odds ratio 80.6; 95% confidence interval 13.4–486.3; $P$ <0.001) and male sex (odds ratio 7.1; 95% confidence interval 1.8–28.1; $P$ = 0.005) were significantly associated with adult rubella. Among men born from 1962 to 1979 (high-risk population, n = 68), conjunctivitis also showed a significant association with adult rubella in the multivariate analysis (odds ratio 24.2; 95% confidence interval 1.1–553.7; $P$ = 0.046) as these patients were not included in the national vaccination program. There was no

**Competing interests:** The authors have declared that no competing interests exist.

difference in the clinical characteristics between one-time vaccination (n = 11) and no vaccination (n = 8) patient in the adult rubella group.

## Conclusions

Conjunctivitis was the key clinical symptom associated with adult rubella. For the early diagnosis of adult rubella, clinicians should focus on assessing conjunctivitis in patients.

## Introduction

Rubella is a contagious, mild viral infection that occurs mostly in children, leading to a vaccine-preventable disease through respiratory droplet [1, 2]. During 2012–2013, Japan had a large rubella outbreak with more than 16000 cases, including 45 cases of congenital rubella syndrome [3]. The Ministry of Health, Labor, and Welfare in Japan issued the Guidelines for the Prevention of Specific Infectious Diseases: Rubella in 2014, and promoted preventive measures throughout the country [4]. However, the second large rubella outbreak has been ongoing since 2018, and about 5000 cases including 3 congenital rubella syndrome were confirmed as at August 2019 [5, 6]. The majority of these outbreaks involved men born between 1962 and 1979 who were not eligible in the national rubella vaccination program for children in Japan [5, 7]. The US Centers for Disease Control and Prevention issued a level 2 travel alert for rubella outbreak in Japan in October 22, 2018; March 11, 2019; and August 7, 2019 [8]. These alerts enhanced precautions so that travelers to Japan could ensure that they were vaccinated against rubella with the measles, mumps, and rubella vaccine before travel. The Global Measles and Rubella Update August 2019 by the World Health Organization revealed Japan as having the second-highest level of rubella reported cases per population [9].

Although rubella in children is characterized by fever, non-confluent maculopapular rash, and lymphadenopathy [2], clinical characteristics are not well described in adult rubella (AR) [10, 11, 12]. There are also no data about the influence of vaccination on the clinical symptoms of AR.

Thus, we conducted a retrospective observational study during two large outbreaks (2012–2013 and 2018–2019), to investigate characteristics of AR, and to evaluate differences in clinical manifestations with/without vaccination.

## Methods

### Ethics statement

This study was approved by the ethics committee of the National Center for Global Health and Medicine (NCGM) (approval no: NCGM-G-003225-00) and was implemented in accordance with the Declaration of Helsinki. Patients' data was anonymized prior to analysis. Due to the retrospective nature of the study, patients' consent was waived.

### Study design and sampling

A retrospective observational study of all symptomatic patients suspected of having rubella, based on clinical symptoms such as fever or rash or lymphadenopathy which are described in the Infectious Disease Surveillance System in Japan [13], was conducted during two outbreaks (January 2012–December 2013 and January 2018–March 2019) at NCGM, Japan. NCGM is a tertiary referral hospital for metropolitan Tokyo and has approximately 780 inpatient beds.

Eligible subjects were those with suspected symptomatic rubella, aged ≥ 18 years who visited NCGM and were screened using rubella-specific IgM test with enzyme immune assay (EIA) kit. The following exclusion criteria were applied: (i) all patients aged < 18 years; (ii) clinically suspected rubella, which resulted in unconfirmed diagnosis due to weak or negative rubella-specific IgM. We defined Japanese men born from 1962 to 1979 as high-risk population because they were not eligible for the national regular rubella vaccination due to the national vaccination program in Japan. The antibody titer for this population was low (about 80%) compared to that of the other generation (over 90%) [14].

## Definition of adult rubella and adult non-rubella

First, we included these study patients with suspected symptomatic rubella based on clinical symptoms such as fever or rash or lymphadenopathy, which were described in the Infectious Disease Surveillance System in Japan [13]. Second, we confirmed the rubella using specific IgM antibodies for rubella in serum and reverse-transcription-polymerase chain reaction (RT-PCR) test. An AR patient was defined as an eligible subject who was confirmed as having rubella on account of the following criteria (based on rubella-specific IgM test using an EIA kit and reverse-transcription-polymerase chain reaction (RT-PCR) test): (i) IgM showing strong positive result with a single serum at first hospital visit; (ii) IgM showing negative or weak result at first hospital visit, but changed to strong positive at follow-up visit; (iii) RT-PCR of throat swab, carried out by the local health government, showing positive rubella. Strong, weak, and negative titers of rubella-specific IgM test, using an EIA kit, were ≥ 1.21, 0.8–1.2, and < 0.8, respectively [15]. Adult non-rubella (ANR) patient was defined as an eligible subject without the evidence of rubella infection.

## Data collection

All eligible subjects who were screened for rubella infection were identified through the hospital laboratory database. The parameters retrieved from patients' records included the following; (i) demographics including age, sex, nationality, pre-exposure to other rubella patients, travel history within last month, pregnancy, number of days from onset to hospital visit; (ii) vaccination status; (iii) rubella-specific IgM serology at first visit; (iv) clinical symptoms including maximum temperature (fever) from onset to the visit; presence and location of rash and lymphadenopathy; conjunctivitis; catarrhal symptoms (cough, pharyngitis, and rhinitis); arthralgia; headache; diarrhea; nausea and vomiting; (v) laboratory tests including complete blood cell counts with atypical lymphocyte, liver enzymes, lactate dehydrogenase (LDH), C-reactive protein (CRP); and (iv) virus subtype. If data was not listed in the electronic medical record, we treated these as missing values, and were removed from the whole number (both numerator and denominator), due to retrospective study.

## Laboratory analysis

The rubella-specific IgM titer was measured by using EIA kit "Seiken" (Denkaseiken, Tokyo, Japan) [15]. The assay protocol, cut-off values, and result interpretations were carried out according to the manufacturer's instruction. The confirmation of rubella and detection of viral genotypes using RT-PCR by the local health government was done on a case-by-case basis until December 2017. Since January 2018, this is now being done according to the pathogen detection manual of the National Institute of Infectious Disease in Japan [16]. Rubella virus gene extraction was performed using real-time RT-PCR. TaqMan RT-PCR and nested RT-PCR have been recommended to local public health centers under the guidance of National Institute of Infectious Disease in Japan [16, 17]. The TaqMan RT-PCR could detect

approximately 90% of throat swab samples that was determined positive by a highly sensitive nested RT-PCR, and was more practical method for rubella laboratory diagnosis. The viral genotypes were determined by a phylogenetic analysis based on the 739-nucleotide window region within rubella virus 1E gene using reported primer sets [18].

## Statistical analysis

Continuous variables were expressed as median with interquartile range (IQR). Categorical variables were shown as absolute and relative frequencies, and compared using the $\chi^2$ test or Fisher's exact test. Mann-Whitney U test was applied for continuous variables. Using logistic regression univariate analysis with odds ratios (OR) and 95% confidence intervals (CI), demographic characteristics and clinical predictive factors between AR and ANR were estimated. Potential predictive factors with a *P* value less than 0.05 in the univariate analysis and *a priori* variables hypothesized to be clinically or epidemiologically important were incorporated into multivariate analysis. The sub-analysis was conducted among high-risk population (Japanese men born from 1962 to 1979) of AR. We also compared the clinical symptoms of AR depending on the vaccination status. Statistical significance was defined as a 2-sided *P*-value of < 0.05. All statistical analyses were performed with SPSS Statistics Version 25 (IBM Corp., Armonk, NY, USA).

## Results

### Description of AR during 2012–2013 and 2018–2019

During the study period, 282 suspected symptomatic rubella patients with screened rubella-specific IgM test results using EIA kit were enrolled. We excluded 61 patients due to the following reasons; (i) patients < 18 years (n = 50); (ii) clinically suspected rubella patients without confirmed diagnosis due to weak or negative rubella-specific IgM at first hospital visit and no paired antibody or RT-PCR (n = 11). Among the remaining 221 patients, 82 were AR and 139 were ANR. The number of strong positive and paired positive of rubella-specific IgM were 49 and 18, respectively. The throat swab rubella RT-PCR result was positive in 15 AR. The causes of infection among ANR were non-rubella viral infections (n = 98) including measles (n = 5), cytomegalovirus infection (n = 4), acute HIV infection (n = 4), Epstein-Barr virus infection (n = 3), chickenpox (n = 3), parvo B19 virus infection (n = 2), dengue fever (n = 1), chikungunya fever (n = 1), drug eruption (n = 20), bacterial infection (n = 10), and others (n = 11). The numbers of AR and ANR were 21 and 33 in 2012, 45 and 60 in 2013, 11 and 28 in 2018, and 5 and 18 in 2019, respectively (Fig 1 and Fig 2).

### Comparison of clinical characteristics between AR and ANR

As shown in Table 1 and Table 2, the median (IQR) age of patients with AR and ANR was 31 (25–41) years and 34 (27–42) years, respectively. The number of AR who received none, one-time, and unknown number of vaccinations were 11 (13.4%), 8 (9.8%), and 63 (76.8%), respectively. Unknown number of vaccinations means that clinician could not confirm the patient's vaccination status. The major symptom found in this study population was rash (100% [82/82] in AR and 87.8% [122/139] in ANR).

At univariate analysis, AR compared to ANR, was significantly associated with male sex (78% vs. 56.1%, OR = 2.8; 95% CI = 1.5–5.2; *P* = 0.001) and pre-exposure to other rubella patients (OR = 4.2; 95% CI = 1.2–14.0; *P* = 0.016). During the two outbreaks, there was significant association with AR during 2012–2013 compared to during 2018–2019 (OR = 2.0; 95% CI = 1.1–3.9; *P* = 0.030). Rash (OR = 1.7; 95% CI = 1.5–1.9; *P* = 0.001), lymphadenopathy

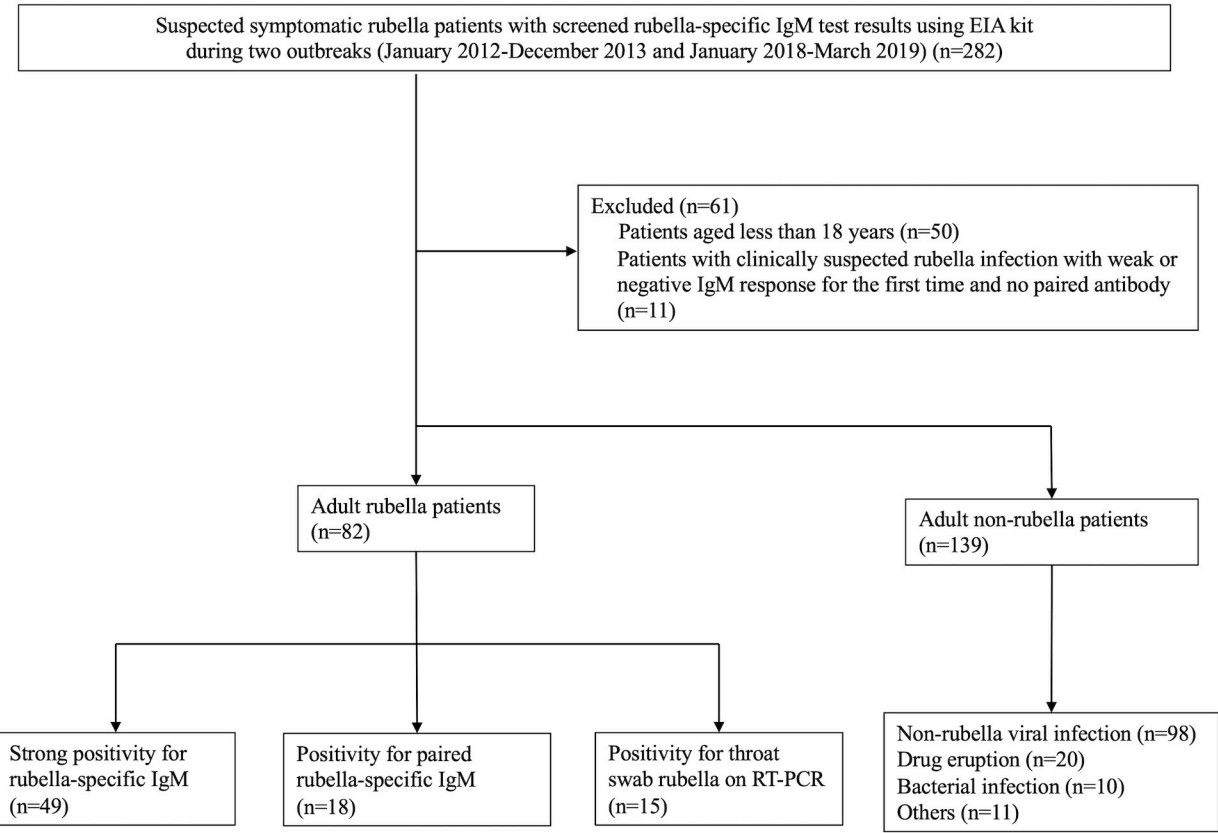

**Fig 1. Flow diagram of study enrollment.** Fig 1 shows the enrollment process of adult rubella (n = 82) and adult non-rubella (n = 139) cases. Rubella-specific IgM was used for the enzyme immune assay (EIA) kit. Strong positive was defined as ≥ 1.21 at first hospital visit. Paired positive was defined as the second strong positive, although the first rubella IgM test was either weak (0.8–1.2) or negative (< 0.8).

(OR = 5.8; 95% CI = 2.8–12.0; $P < 0.001$), conjunctivitis (OR = 66.7; 95% CI = 22.3–199.7; $P < 0.001$), catarrhal symptoms (OR = 2.2; 95% CI = 1.2–3.8; $P = 0.007$), and arthralgia (OR = 1.9; 95% CI = 1.0–3.6; $P = 0.039$) were more common in AR compared to ANR. In AR, there was significantly increased median [IQR] LDH and decreased median [IQR] white blood cell (WBC), platelet, and CRP (Table 1). Multivariate analysis showed that conjunctivitis (OR = 80.6; 95% CI = 13.4–486.3; $P < 0.001$) and male sex (OR = 7.1; 95% CI = 1.8–28.1; $P = 0.005$) were significantly associated with AR. Of 33 AR observed during 2018–2019, the majority with confirmed virus genotype showed genotype 1E (n = 14), and only one patient who seemed to have been infected in India had genotype 2B. Virus genotype during 2012–2013 was not confirmed because the local health government was not evaluating the subtype at that time (Table 1 and Table 2).

## Clinical characteristics among high-risk population of AR

Among the high-risk population (n = 68), in univariate analysis, face rash, cervical lymphade-nopathy, conjunctivitis, catarrhal symptoms, decreased WBC, and CRP were significantly observed in AR. Conjunctivitis was significantly associated with AR in multivariate analysis (OR = 24.2; 95% CI = 1.1–553.7; $P = 0.046$) (Table 3). Among none (n = 11) and one-time (n = 8) vaccination times in AR, at univariate analysis, no difference was shown in the demographic characteristics, clinical symptoms, and laboratory results (Table 4).

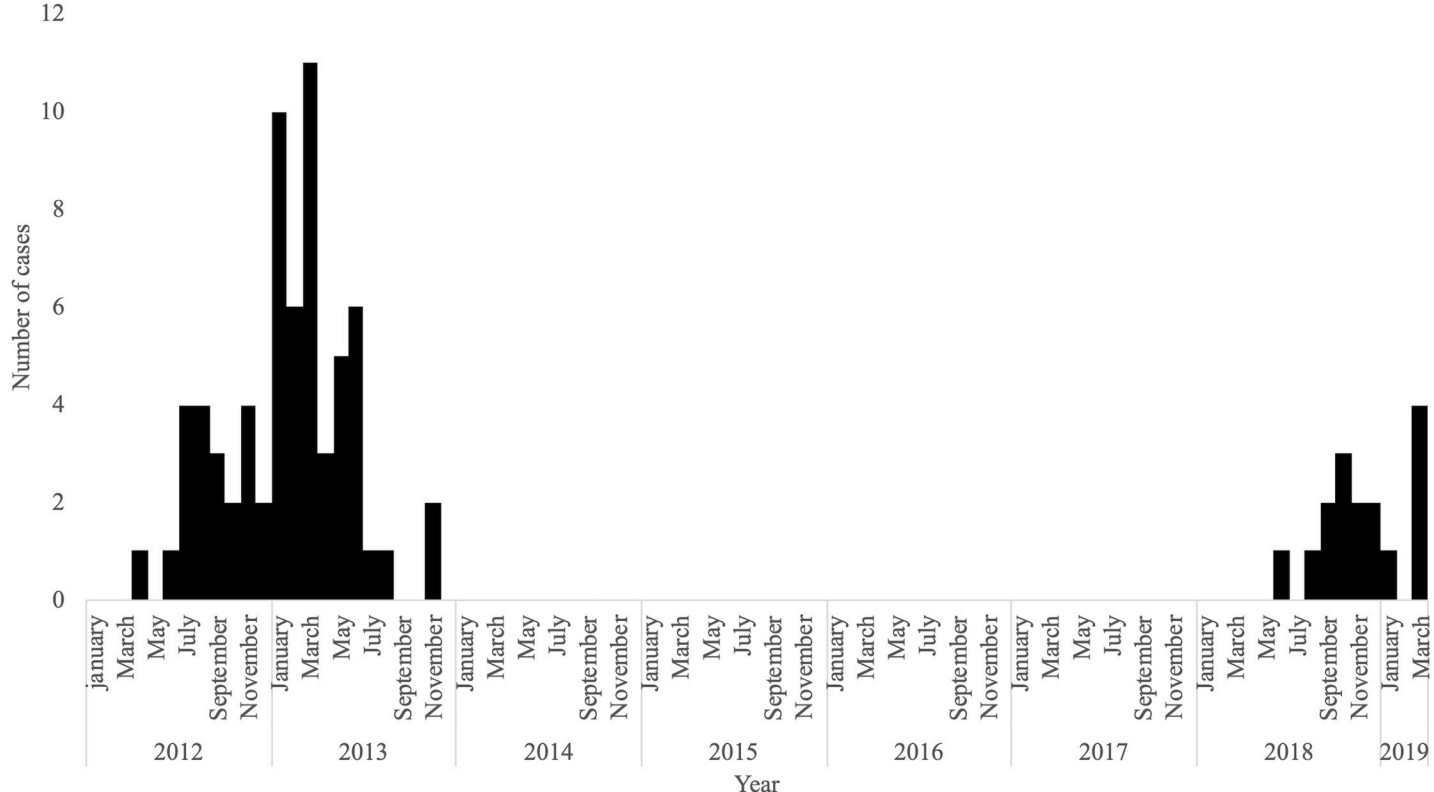

**Fig 2. Number of adult rubella cases from January 2012 to December 2013 and January 2018 to March 2019, n = 82.** Fig 2 shows the number of adult rubella cases during the study period.

**Table 1. Univariate and multivariate analysis of backgrounds of adult rubella, n = 221.**

| Category | Variable | Adult rubella | | Adult non-rubella | | Univariate analysis | | | Multivariate analysis | | |
|---|---|---|---|---|---|---|---|---|---|---|---|
| | | (n = 82, 36.8%) | | (n = 139, 63.2%) | | OR | (95% CI) | P value | OR | (95% CI) | P value |
| Demographic characteristics | Age, median (IQR), years | 31 | (25–41) | 34 | (27–42) | | | 0.218 | 1.0 | (0.9–1.0) | 0.137 |
| | Male sex | 64 | (78.0) | 78 | (56.1) | 2.8 | (1.5–5.2) | 0.001 | 7.1 | (1.8–28.1) | 0.005 |
| | Japanese | 81 | (98.8) | 131 | (94.2) | 4.9 | (0.6–40.3) | 0.159 | | | |
| | Pregnancy | 0 | (0.0) | 2 | (1.4) | 0.6 | (0.6–0.7) | 0.392 | | | |
| | From 2012 to 2013 | 66 | (80.5) | 93 | (66.9) | 2.0 | (1.1–3.9) | 0.030 | 0.476 | (0.1–2.3) | 0.351 |
| | Pre-exposure to other rubella patients | 9 | (11.0) | 4 | (2.9) | 4.2 | (1.2–14.0) | 0.016 | | | |
| | Travel history | 6 | (7.3) | 29 | (20.9) | 0.3 | (0.1–0.8) | 0.008 | | | |
| | Number of days from onset to hospital visit | 4 | (3–7) | 5 | (3–9) | | | 0.622 | | | |
| Vaccination | None | 11 | (13.4) | 9 | (6.5) | 2.2 | (0.9–5.7) | 0.082 | | | |
| | One time | 8 | (9.8) | 13 | (9.6) | 1.0 | (0.4–2.6) | 0.921 | | | |
| | Two times | 0 | (0.0) | 6 | (4.4) | 0.6 | (0.6–0.7) | 0.059 | | | |
| | Unknown | 63 | (76.8) | 111 | (79.9) | 0.8 | (0.4–1.6) | 0.595 | | | |

Unless otherwise stated, data are presented as n (%)

Continuous variable data are presented as median (IQR)

OR; odds ratio, CI; confidence interval, IQR; interquartile range

**Table 2. Univariate and multivariate analysis of clinical characteristics and laboratory findings of adult rubella, n = 221.**

| Category | Variable | Adult rubella | | Adult non-rubella | | Univariate analysis | | | Multivariate analysis | | |
|---|---|---|---|---|---|---|---|---|---|---|---|
| | | (n = 82, 36.8%) | | (n = 139, 63.2%) | | OR | (95% CI) | P value | OR | (95% CI) | P value |
| Clinical symptoms | Maximum temperature (fever) from onset to hospital visit, ˚C | 38.0 | (37.3–39.0) | 38.1 | (37.3–39.0) | | | 0.673 | | | |
| | Rash | 82 | (100.0) | 122 | (87.8) | 1.7 | (1.5–1.9) | 0.001 | | | |
| | Face | 64 | (87.7) | 48 | (41.0) | 10.2 | (4.6–22.5) | <0.001 | 4.3 | (0.9–19.7) | 0.060 |
| | Trunk | 81 | (98.8) | 108 | (90.0) | 9.0 | (1.1–70.6) | 0.016 | | | |
| | Extremity | 82 | (100) | 110 | (93.2) | 1.7 | (1.5–2.0) | 0.013 | | | |
| | Lymphadenopathy | 66 | (85.7) | 64 | (50.8) | 5.8 | (2.8–12.0) | <0.001 | | | |
| | Cervical | 61 | (80.3) | 62 | (49.6) | 4.1 | (2.1–8.0) | <0.001 | 2.0 | 0.5–7.9 | 0.327 |
| | Peri-auricular | 30 | (57.7) | 8 | (6.6) | 19.3 | (7.8–47.6) | <0.001 | | | |
| | Conjunctivitis | 68 | (94.4) | 26 | (20.3) | 66.7 | (22.3–199.7) | <0.001 | 85.6 | (14.2–514.0) | <0.001 |
| | Catarrhal symptoms* | 47 | (61.0) | 57 | (41.9) | 2.2 | (1.2–3.8) | 0.007 | 2.8 | (0.8–10.0) | 0.116 |
| | Arthralgia | 26 | (31.7) | 27 | (19.4) | 1.9 | (1.0–3.6) | 0.039 | | | |
| | Headache | 20 | (24.4) | 32 | (23.0) | 1.1 | (0.6–2.0) | 0.817 | | | |
| | Diarrhea | 10 | (12.2) | 16 | (11.5) | 1.1 | (0.5–2.5) | 0.879 | | | |
| | Nausea or vomiting | 9 | (11.0) | 9 | (6.5) | 1.8 | (0.7–4.7) | 0.237 | | | |
| Laboratory test | WBC, /μL | 4710 | (3290–6010) | 6100 | (3620–7572) | | | 0.012 | 1.0 | (1.0–1.0) | 0.003 |
| | Atypical lymphocyte, /μL | 69 | (24–174) | 0 | (0–0) | | | <0.001 | | | |
| | Platelet×$10^4$, /μL | 15.5 | (13.6–18.2) | 19.8 | (14.2–23.8) | | | <0.001 | | | |
| | AST, U/L | 34 | (27–44) | 28 | (19–46) | | | 0.013 | | | |
| | ALT, U/L | 31 | (18–47) | 28 | (17–54) | | | 0.630 | | | |
| | LDH, U/L | 300 | (231–367) | 225 | (185–292) | | | <0.001 | | | |
| | CRP, mg/dL | 0.7 | (0.3–1.8) | 1.3 | (0.3–3.6) | | | 0.030 | 0.7 | (0.5–1.0) | 0.039 |
| Rubella-specific IgM serology at first hospital visit | Strong positive | 59 | (72.0) | 0 | (0.0) | 0.1 | (0.1–0.2) | <0.001 | | | |
| | Weak positive | 5 | (6.1) | 1 | (0.7) | 9.0 | (1.0–78.1) | 0.027 | | | |
| | Negative | 18 | (22.0) | 138 | (99.3) | 0.002 | (0.001–0.02) | <0.001 | | | |
| Virus subtype | 1E | 14 | (17.1) | | | | | | | | |
| | 2B | 1 | (1.2) | | | | | | | | |
| | Unknown | 68 | (82.9) | | | | | | | | |

Unless otherwise stated, data are presented as n (%)

Continuous variable data are presented as median (IQR)

OR; odds ratio, CI; confidence interval, IQR; interquartile range, WBC; white blood cell, AST; aspartate aminotransferase

ALT; alanine aminotransferase, LDH; lactate dehydrogenase, CRP; C-reactive protein

*Catarrhal symptoms were defined as one of cough, pharyngitis and rhinitis.

**Table 3. Multivariate analysis of the characteristics among high-risk population\* of adult rubella, n = 68.**

| Variable | Adult rubella | | Adult non-rubella | | Univariate analysis | | | Multivariate analysis | | |
|---|---|---|---|---|---|---|---|---|---|---|
| | (n = 26, 38.2%) | | (n = 42, 61.8%) | | OR | 95% CI | P value | OR | 95% CI | P value |
| Age, years | 42 | (37.0–44.3) | 42 | (35.8–44.0) | | | 0.622 | 1.1 | (0.9–1.4) | 0.360 |
| From 2012 to 2013 | 22 | (84.6) | 28 | (66.7) | 2.6 | (0.8–9.5) | 0.103 | | | |
| Face rash | 19 | (82.6) | 10 | (31.3) | 10.5 | (2.8–38.8) | <0.001 | 9.3 | (0.6–143.1) | 0.111 |
| Cervical lymphadenopathy | 16 | (66.7) | 15 | (38.5) | 3.2 | (1.1–9.3) | 0.030 | 5.2 | (0.3–97.3) | 0.271 |
| Conjunctivitis | 21 | (91.3) | 7 | (18.9) | 45.0 | (8.5–238.4) | <0.001 | 24.2 | (1.1–553.7) | 0.046 |
| Catarrhal symptoms† | 17 | (68.0) | 8.0 | 8.0 | 3.0 | (1.1–8.5) | 0.036 | 8.0 | (0.4–156.9) | 0.172 |
| WBC, /μL | 4775 | (2998–6138) | 6070 | (3530–9100) | | | 0.038 | 1.0 | (1.0–1.0) | 0.123 |
| CRP, mg/dL | 0.6 | (0.4–1.6) | 1.8 | (0.6–3.9) | | | 0.020 | 0.7 | (0.3–1.5) | 0.332 |

Unless otherwise stated, data are presented as n (%)

Continuous variable data are presented as median (IQR)

OR; odds ratio, CI; confidence interval, IQR; interquartile range, WBC; white blood cell, CRP; C-reactive protein

\*High risk population were defined as men born from 1962 to 1979.

†Catarrhal symptoms were defined as one of cough, pharyngitis and rhinitis.

## Discussion

This study showed that conjunctivitis, with a 66.7-fold (22.3–199.7) likelihood, was the most clinical predictive factor of AR, among clinically suspected rubella. The three classical clinical manifestations of rubella among children were reported as fever, rash, and lymphadenopathy, but our findings showed that rash and lymphadenopathy were significantly associated with AR in univariate analysis only. Limited clinical studies have evaluated for conjunctivitis among AR. The only past case series of rubella outbreak during 2012–2013 in Japan reported that conjunctivitis was observed in 77.8% of adult patients [10]. We think that rubella produces follicular reaction including follicular conjunctivitis along with catarrhal symptoms [19, 20].

Male sex and age were significantly associated with AR in univariate analysis, but only male sex remained associated with AR in multivariate analysis (OR = 7.1; 95% CI = 1.8–28.1; P = 0.005). Japanese men born between 1962 and 1979 were regarded as high-risk rubella population group due to the non-inclusion in the national vaccination program in Japan [14]. From August 1977 to March 1995, the single-dose rubella vaccine was given to junior high school women through the national immunization program. The program was extended for universal coverage, which included men from April 1995, meaning that Japanese men who graduated from junior high school then did not have an opportunity to receive rubella vaccine through the regular national vaccination program. The antibody level among these high-risk population was low (about 80%), compared to that of the other populations (over 90%) [7]. Sub-analysis restricted to the high-risk group population showed that conjunctivitis was also a crucial finding (OR = 24.2; 95% CI = 1.1–553.7).

There were no differences in clinical characteristics between AR who received one-time vaccination and unvaccinated patients. No data reflected symptoms of rubella in both children and adults pertaining to the vaccination status. Rubella vaccine was considered as highly immunogenic [21, 22], but 8 (9.8%) AR patients had received one-time vaccination in our study. Otherwise, no two-time vaccination was observed in our study. Although further information about vaccination and immunization status among patients with unknown vaccination were not collected due to retrospective research, our study showed that not only one- but two-time vaccinations were needed to prevent rubella. To stop the ongoing rubella outbreak,

**Table 4. Univariate analysis of the characteristics among adult rubella patients between none and one-time vaccination, n = 19.**

| Category | Variable | None | | One-time | | Univariate analysis | | |
|---|---|---|---|---|---|---|---|---|
| | | (n = 11, 57.9%) | | (n = 8, 42.1%) | | OR | (95% CI) | *P* value |
| Demographic characteristics | Age, years (median, IQR) | 28 | (23–45) | 27 | (22–36) | | | 0.492 |
| | Male sex | 5 | (45.5) | 7 | (87.5) | 0.1 | (0.01–1.3) | 0.080 |
| | Japanese | 11 | (100) | 8 | (100) | | | |
| | Pregnancy | 0 | (0.0) | 0 | (0.0) | | | |
| | From 2012 to 2013 | 6 | (54.5) | 7 | (87.5) | 0.2 | (0.02–1.9) | 0.153 |
| | Pre-exposure to other rubella patients | 1 | (9.1) | 0 | (0.0) | 0.6 | (0.4–0.8) | 0.579 |
| | Travel history | 1 | (9.1) | 0 | (0.0) | 0.6 | (0.4–0.8) | 0.579 |
| | Number of days from onset to hospital visit | 5 | (2–11) | 7 | (3–9) | | | 0.903 |
| Rubella-specific IgM serology at first hospital visit | Strong positive | 6 | (54.5) | 7 | (87.5) | 0.2 | (0.02–1.9) | 0.153 |
| | Weak positive | 2 | (18.2) | 0 | (0.0) | 0.5 | (0.3–0.8) | 0.322 |
| | Negative | 3 | (27.3) | 1 | (12.5) | 2.6 | (0.2–31.3) | 0.426 |
| Clinical symptom | Maximum temperature (fever) from onset to hospital visit, ˚C | 37.4 | (37.1–38.0) | 38.0 | (37.5–39.3) | | | 0.179 |
| | Rash | 11 | (100.0) | 8 | (100.0) | | | |
| | Face | 8 | (72.7) | 7 | (87.5) | 0.4 | (0.03–4.6) | 0.426 |
| | Trunk | 10 | (90.9) | 8 | (100.0) | 1.8 | (1.2–2.7) | 0.579 |
| | Extremity | 11 | (100.0) | 8 | (100.0) | | | |
| | Lymphadenopathy | 10 | (90.9) | 7 | (87.5) | 1.4 | (0.1–28.9) | 0.678 |
| | Cervical | 10 | (90.9) | 5 | (62.5) | 6.0 | (0.5–73.5 | 0.177 |
| | Peri-auricular | 8 | (88.9) | 3 | (50.0) | 8.0 | (0.6–110.3) | 0.143 |
| | Conjunctivitis | 11 | (100.0) | 5 | (83.3) | 3.2 | (1.5–6.6) | 0.353 |
| | Catarrhal symptoms* | 3 | (27.3) | 6 | (75.0) | 0.1 | (0.02–01.0) | 0.055 |
| | Arthralgia | 3 | (27.3) | 4 | (50.0) | 0.4 | (0.1–2.6) | 0.297 |
| | Headache | 2 | (18.2) | 4 | (50.0) | 0.2 | (0.03–1.8) | 0.166 |
| | Diarrhea | 3 | (27.3) | 1 | (12.5) | 2.6 | (0.2–31.3) | 0.426 |
| | Nausea or vomiting | 2 | (18.2) | 1 | (12.5) | 1.6 | (0.1–20.9) | 0.624 |
| Laboratory test | WBC, /μL | 3540 | (2938–4818) | 5370 | (3548–6320) | | | 0.083 |
| | Atypical lymphocyte, /μL | 57 | (34–118) | 75 | (30–171) | | | 0.805 |
| | Platelet×$10^4$, /μL | 17.5 | (14.7–19.7) | 14.3 | (13.1–19.4) | | | 0.460 |
| | AST, U/L | 34 | (26–41) | 43 | (30–65) | | | 0.173 |
| | ALT, U/L | 31 | (14–40) | 38 | (23–107) | | | 0.122 |
| | LDH, U/L | 249 | (202–288) | 300 | (271–365) | | | 0.074 |
| | CRP, mg/dL | 0.4 | (0.2–0.5) | 0.6 | (0.4–1.8) | | | 0.203 |
| Virus subtype | 1E | 5 | (45.5) | 1 | (5.3) | 5.8 | (0.5–64.8) | 0.177 |
| | Unknown | 6 | (54.5) | 7 | (87.5) | 0.2 | (0.02–1.91) | 0.177 |

Unless otherwise stated, data are presented as n (%)

Continuous variable data are presented as median (IQR)

OR; odds ratio, CI; confidence interval, IQR; interquartile range, WBC; white blood cell, AST; aspartate aminotransferase

ALT; alanine aminotransferase, LDH; lactate dehydrogenase, CRP; C-reactive protein

*Catarrhal symptoms were defined as one of cough, pharyngitis and rhinitis.

Japanese government started an evaluation of an antibody test and catch-up vaccination program for high-risk population since April 2019 [23].

This study has several limitations. First, both AR and ANR in this study excluded the asymptomatic patients, who are estimated to have up to 50% of rubella cases [19]. However, the influence of the asymptomatic rubella infection is not well known. Our main purpose in this study was also to determine the clinical characteristics of AR to enhance early diagnosis. Second, several patients (76.8% AR and 79.9% ANR) did not know their vaccination history. In Japan, vaccination status was recorded in the mother and child health paper handbook. This handbook was personally kept, and if patients did not bring this handbook during the hospital visit, clinicians did not refer to the vaccination status. Third, 81.7% AR and 37.4% ANR who were diagnosed with rubella-specific IgM test only without PCR might be misclassified [24, 25]. PCR-based rubella diagnosis is reliable, but no commercial diagnostic PCR tests could be performed in Japan, and the local health government only conducted RT-PCR for selected cases before 2018. However, currently available commercial rubella-specific IgM test (EIA kit), which was used in this study has high specificity of $\geq$ 95% [26]. The possibility of misclassification of false-positive rubella IgM was thought to be low. Moreover, previous reports showed that this rubella-specific IgM test (EIA kit) indicated highly positive results for rubella (reaching 80%) among patients who received this test after 5 days of the onset of symptoms [27]. Therefore, we conducted the multivariate analysis between 33 AR and 72 ANR who were diagnosed after 5 days of onset of symptoms or confirmed by RT-PCR. Based on the results of univariate analysis and the stability of the model, we conducted the multivariate analysis adjusted for age, sex, cervical lymphadenopathy, and conjunctivitis, and only conjunctivitis remained significantly associated with AR (OR = 61.8; 95% CI = 7.2–528.8; $P$ <0.001), similar to the full analysis. Fourth, we did not evaluate difference in conjunctivitis between AR and measles due to the low number of measles cases (n = 5). Conjunctivitis has been reported as one of the major clinical characteristics of measles [28]; however, only 2 of 5 measles cases showed conjunctivitis in our study. There is the possibility that the clinical symptoms were underestimated due to the retrospective nature of the study. To evaluate the clinical characteristics between AR and measles, further study is needed. Finally, the inclusion criteria of these study patients (AR and ANR) were suspected symptomatic rubella patients based on clinical symptoms such as fever or rash or lymphadenopathy, which were described in the Infectious Disease Surveillance System in Japan, and they were evaluated rubella infection using laboratory examination, including RT-PCR and rubella IgM antibody test. Therefore, rubella-suspected symptoms such as lymphadenopathy were likely to be pre-selected bias in this research. However, our new finding "conjunctivitis, the key clinical characteristic of adult rubella" is thought to be important to distinguish diseases within suspected symptomatic rubella patients at clinical practice.

In conclusion, our study is the first clinical study globally to indicate the association between conjunctivitis and AR among clinically suspected rubella patients. During rubella outbreak, we believe that clinicians need to pay careful attention to the occurrence of conjunctivitis in addition to the three classical symptoms (fever, rash, and lymphadenopathy) of AR, for an early diagnosis. With the upcoming Tokyo Olympic/Paralympic Games in 2020, a major global event with a potentially unprecedented number of visitors entering Japan; continued rubella diagnosis, prevention, and control will be important.

## Supporting information

**S1 Table. A data sheet of the present study.**
(XLSX)

## Acknowledgments

We thank all the clinical staff at the NCGM for their dedicated clinical practice and patient care.

## Author Contributions

**Conceptualization:** Hidetoshi Nomoto, Masahiro Ishikane.

**Data curation:** Hidetoshi Nomoto.

**Investigation:** Hidetoshi Nomoto.

**Methodology:** Hidetoshi Nomoto, Masahiro Ishikane.

**Resources:** Takato Nakamoto, Masayuki Ohta, Shinichiro Morioka, Kei Yamamoto, Satoshi Kutsuna, Shunsuke Tezuka, Junwa Kunimatsu.

**Supervision:** Masahiro Ishikane, Norio Ohmagari.

**Validation:** Masahiro Ishikane.

**Writing – original draft:** Hidetoshi Nomoto.

**Writing – review & editing:** Hidetoshi Nomoto, Masahiro Ishikane, Norio Ohmagari.

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
