## [Decision Letter · Decision Letter 0]

14 Feb 2020

PONE-D-19-34587

Clinical characteristics of adult rubella in Japan during two large outbreaks, 2012-2013 and 2018-2019

PLOS ONE

Dear Dr Ishikane,

Thank you for submitting your manuscript to PLOS ONE. After careful consideration, we feel that it has merit but does not fully meet PLOS ONE’s publication criteria as it currently stands. Therefore, we invite you to submit a revised version of the manuscript that addresses the points raised during the review process.

Your manuscript has been reviewed by two experts in the field. Both reviewers underscored the necessity to better describe the cohort used in your study. Although the limitation of the study are mentioned in the discussion a critical analysis of the cohort and the bias that can affect the conclusion should be made explicit. Please revise the data presented in the tables to address the concerns of the reviewers.

I personally encourage the authors to modify the title of the manuscript as suggested by reviewer 1.

We would appreciate receiving your revised manuscript by Mar 30 2020 11:59PM. To enhance the reproducibility of your results, we recommend that if applicable you deposit your laboratory protocols in protocols.io, where a protocol can be assigned its own identifier (DOI) such that it can be cited independently in the future. For instructions see: http://journals.plos.org/plosone/s/submission-guidelines#loc-laboratory-protocols

We look forward to receiving your revised manuscript.

Kind regards,

Giovanna Barba-Spaeth, Ph.D.

Academic Editor

PLOS ONE

Journal Requirements:

2. Please upload a copy of Supporting Information Table 1 which you refer to in your text on page 34.

Reviewers' comments:

Reviewer's Responses to Questions

**Comments to the Author**

1. Is the manuscript technically sound, and do the data support the conclusions?

Reviewer #1: Yes

Reviewer #2: Partly

2. Has the statistical analysis been performed appropriately and rigorously? 

Reviewer #1: Yes

Reviewer #2: I Don't Know

3. Have the authors made all data underlying the findings in their manuscript fully available?

Reviewer #1: Yes

Reviewer #2: Yes

4. Is the manuscript presented in an intelligible fashion and written in standard English?

Reviewer #1: Yes

Reviewer #2: Yes

5. Review Comments to the Author

Reviewer #1: 1. Title

According to your results and conclusion, I think that the title could be: “Conjunctivitis, the main clinical characteristic of adult rubella in Japan during two large outbreaks, 2012–2013 and 2018–2019”, incorporating your most significant results.

2. Methods section.

a. Please clarify the exclusion criteria and if RT PCR for rubella was obtained and was negative in patients with weak or negative antibodies type IgM for rubella.

b. About RT PCR, please give more information regarding the detection limit of method and about the kit used for extraction.

c. Page 12, line 201, please rephrase and clarify about unknown number of vaccinations.

d. Page 19, line 223, you present genotype 1E, as the most common. Nevertheless, you do not describe genotyping in the methods section.

e. Please clarify the high-risk population group in the section: “study design and sampling”.

3. Results/Tables

a. Table 1. Not easy to read. More readable if you divide it in 2 tables.

b. The same information is repeated in the manuscript, could be removed from the tables or from the text.

c. Please add as a new table the comparative results, with statistical analysis, of the two periods of your study.

4. It is very interesting the main finding of conjunctivitis in AR. Is it possible to conduct a pathogenetic explanation? In conjunction with congenital rubella, where conjunctivitis is also a main clinical symptom.

5. Regarding your result of not known vaccinations, it could be added in the discussion section that a further investigation, retrospectively, of the immunization status of this subjects, would be very informative about the need for two dose vaccination scheme.

6. A few spelling and grammatical suggestion

a. Line 33 add comma after kit

b. Line 81 change with to by

c. Line 127 add comma before and after “using an EIA kit”

d. Line 135 replace one with last

e. Line 138 remove defend as

f. Line 139 and nausea change to: nausea;

g. Line 140 change test and enzyme to tests and enzymes respectively

Reviewer #2: Ishikane reported a comprehensive clinical information about 82 cases of Rubella virus infection in Japan during 2012-2013 and 2018-2019. Rubella outbreak is a very important health threatening event, the data provided in this MS can help physicians to understand the phenotype of Rubella infection and might improve the clinical diagnosis. Generally, the MS is well-written.

However, I have three major concern.

1. In this MS, the statistic is by comparing the AR with ANR; however, it is not well defined how these patients (AR+ANR) were selected. This information is very important and might create a bias for the statistic, authors should explain how these cohort were selected.

2. Follow the same idea, I am not sure if it is sufficient only compared the AR and ANR can draw a conclusion. As it is well know, Lymphadenopathy is a key phenotype of Rubella infection, indeed, this phenotype is common in both AR and ANR group, this suggested that lymphadenopathy is being considered and therefore these patients were further received the Rubella screening. This will also cause the bias; if authors only compared AR and ANR groups, authors should emphasis this point through the MS that this is pre-selected cohort analysis.

3. I am very confused about the data presented in Tables. For example, in table I, Trunk 81 (98.8%); extremity 82 (100%); but conjuctivitis 68 (94.4%) and peri-auricular 30 (57.7%). Could authors explain this point? this is extremely important, since this will large affect the conclusion. I suggest that 94.4% is 68 cases have conjuctivitis and 4 cases did not. it means that only 72 in total instead of 82. I don't know why 10 cases were exclude?

6. PLOS authors have the option to publish the peer review history of their article (what does this mean?). If published, this will include your full peer review and any attached files.

Reviewer #1: No

Reviewer #2: No

---

## [Author Response · Author response to Decision Letter 0]

20 Mar 2020

Dr Joerg Heber

Editor-in-Chief

PLOS ONE

March 16, 2020

Re: PONE-D-19-34587

Title “Conjunctivitis, the key clinical characteristic of adult rubella in Japan during two large outbreaks, 2012–2013 and 2018–2019” Authors: Nomoto H et al.

Dear Dr. Heber,

Thank you for your e-mail on February 14, 2020. We were pleased to read the positive evaluation of our manuscript and its potential acceptance for publication in PLOS ONE, subject to adequate revision and responses to the Reviewers’ comments.

Based on your instruction, we logged into Editorial Manger website and submitted the two files of the revised manuscript (one clean and one with the track changes). 

Please find attached point-by-point response to the comments raised by the Reviewers. 

We take this opportunity to express our gratitude to the Reviewers for the constructive and useful remarks. The comments allowed us to identify areas in our manuscript that needed modification and clarification. We also thank you for allowing us to resubmit a revised copy of the manuscript.

I hope that the revised manuscript is now acceptable for publication in PLOS ONE.

Sincerely Yours,

Masahiro Ishikane, M.D., Ph.D. (Corresponding author)

Department of Disease Control and Prevention Center, National Center for Global Health and Medicine

1-21-1 Toyama, Shinjuku-ku, Tokyo, 162-8655, Japan. 

Tel: +81-3-3202-7181

Fax: +81-3-3202-1012 

E-mail: ishikanemasahiro@gmail.com

Reviewer #1

1. Title

According to your results and conclusion, I think that the title could be: “Conjunctivitis, the main clinical characteristic of adult rubella in Japan during two large outbreaks, 2012–2013 and 2018–2019”, incorporating your most significant results.

Thank you for your suggestion. We changed our title that is more representative of the result and conclusion

(Before)

Title “Clinical characteristics of adult rubella in Japan during two large outbreaks, 2012-2013 and 2018-2019”

(After)

Title “Conjunctivitis, the key clinical characteristic of adult rubella in Japan during two large outbreaks, 2012–2013 and 2018–2019”

2. Methods section.

a. Please clarify the exclusion criteria and if RT PCR for rubella was obtained and was negative in patients with weak or negative antibodies type IgM for rubella.

Thank you for your advice. First, we included these study patients with suspected symptomatic rubella based on clinical symptoms such as fever or rash or lymphadenopathy which are described in the Infectious Disease Surveillance System in Japan [13]. Second, we confirmed the rubella using specific IgM antibodies for rubella in serum and RT-PCR. RT-PCR is gold standard to diagnose rubella and specific IgM antibodies in serum is detection within a first few days also suggest acute rubella infection [27. Because RT-PCR was considered useful as gold standard to confirm rubella, if RT-PCR was positive, the patient was considered rubella regardless of IgM result. If the result of RT-PCR for rubella was negative with weak or negative antibodies type IgM for rubella at first hospital visit, we decided on changing or not to strong positive at follow-up visit. We also had concerns about misclassifying patients early in onset. Therefore, we mentioned it in limitation section, and sub-analysis was also performed in population 5 days or more after onset. According to your suggestion, we added the additional explanation about exclusion criteria about adult rubella and adult non-rubella, in method part. Also, we added new reference about Infectious Disease Surveillance System in Japan [13].

(Before: Study design and sampling)

A retrospective observational study of all symptomatic patients suspected of having rubella was conducted during two outbreaks (January 2012–December 2013 and January 2018–March 2019) at NCGM, Japan.

(After: Study design and sampling)

A retrospective observational study of all symptomatic patients suspected of having rubella, based on clinical symptoms such as fever or rash or lymphadenopathy which are described in the Infectious Disease Surveillance System in Japan [13], was conducted during two outbreaks (January 2012–December 2013 and January 2018–March 2019) at NCGM, Japan.

(Before: Definition of adult rubella and adult non-rubella)

An AR patient was defined as an eligible subject who was confirmed as having rubella on account of the following criteria (based on rubella-specific IgM test using an EIA kit and reverse-transcription-polymerase chain reaction (RT-PCR) test): (ⅰ) IgM showing strong positive result with a single serum at first hospital visit; (ⅱ) IgM showing negative or weak result at first hospital visit, but changed to strong positive at follow-up visit; (ⅲ) RT-PCR of throat swab, carried out by the local health government, showing positive rubella. Strong, weak, and negative titers of rubella-specific IgM test using an EIA kit were ≥ 1.21, 0.8–1.2, and < 0.8, respectively [13]. Adult non-rubella (ANR) patient was defined as an eligible subject without the evidence of rubella infection.

(After: Definition of adult rubella and adult non-rubella) 

First, we included these study patients with suspected symptomatic rubella based on clinical symptoms such as fever or rash or lymphadenopathy which are described in the Infectious Disease Surveillance System in Japan [13]. Second, we confirmed the rubella using specific IgM antibodies for rubella in serum and reverse-transcription-polymerase chain reaction (RT-PCR) test. An AR patient was defined as an eligible subject who was confirmed as having rubella on account of the following criteria: (i) IgM showing strong positive result with a single serum at first hospital visit; (ii) IgM showing negative or weak result at first hospital visit, but changed to strong positive at follow-up visit; (iii) RT-PCR of throat swab, carried out by the local health government, showing positive rubella. Strong, weak, and negative titers of rubella-specific IgM test using an EIA kit were ≥ 1.21, 0.8–1.2, and < 0.8, respectively [15]. ANR patient was defined as an eligible subject without the evidence of rubella infection.

13. National Institute of Infectious Disease in Japan. Infectious Disease Surveillance System in Japan. [cited 2020 March 14]. Available from: https://www.niid.go.jp/niid/images/epi/nesid/nesid_en.pdf

b. About RT PCR, please give more information regarding the detection limit of method and about the kit used for extraction.

Thank you for your suggestion. Rubella virus gene extraction was performed using real-time RT-PCR. TaqMan RT-PCR and nested RT-PCR have been recommended to local public health centers under the guidance of National Institute of Infectious Disease in Japan [16]. The TaqMan RT-PCR could detect approximately 90% of throat swab samples that was determined positive by a highly sensitive nested RT-PCR, and was more practical method for rubella laboratory diagnosis. We added more information about RT-PCR with new reference [16] in method section.

(Before)

The rubella-specific IgM titer was measured by using EIA kit “Seiken” (Denkaseiken, Tokyo, Japan) [13]. The assay protocol, cut-off values, and result interpretations were carried out according to the manufacturer's instruction. The confirmation of rubella and detection of viral genotypes using RT-PCR by the local health government was done on a case-by-case basis until December 2017. Since January 2018, this is now being done according to the pathogen detection manual of the National Institute of Infectious Disease in Japan [14]. The viral genotypes were determined by a phylogenetic analysis based on the 739-nucleotide window region within rubella virus 1E gene using reported primer sets [15].

(After)

The rubella-specific IgM titer was measured by using EIA kit “Seiken” (Denkaseiken, Tokyo, Japan) [15]. The assay protocol, cut-off values, and result interpretations were carried out according to the manufacturer's instruction. The confirmation of rubella and detection of viral genotypes using RT-PCR by the local health government was done on a case-by-case basis until December 2017. Since January 2018, this is now being done according to the pathogen detection manual of the National Institute of Infectious Disease in Japan [16]. Rubella virus gene extraction was performed using real-time RT-PCR. TaqMan RT-PCR and nested RT-PCR have been recommended to local public health centers under the guidance of National Institute of Infectious Disease in Japan [16, 17]. The TaqMan RT-PCR could detect approximately 90% of throat swab samples that was determined positive by a highly sensitive nested RT-PCR, and was more practical method for rubella laboratory diagnosis. The viral genotypes were determined by a phylogenetic analysis based on the 739-nucleotide window region within rubella virus 1E gene using reported primer sets [18].

14.　National Institute of Infectious Diseases in Japan. The pathogen detection manual. [cited 2019 Aug 15]. Available from: https://www.niid.go.jp/niid/ja/labo-manual.html#class5.

16. Okamoto K, Mori Y, Komagome R, Nagano H, Miyoshi M, Okano M, et al. Evaluation of Sensitivity of TaqMan RT-PCR for Rubella Virus Detection in Clinical Specimens. J Clin Virol. 2016;80: 98-101.

c. Page 12, line 201, please rephrase and clarify about unknown number of vaccinations.

Thank you for your advice. "Unknown number of vaccinations" means that clinician could not confirm the patient's vaccination status. As we mentioned in the limitation, vaccination status was recorded in the mother and child health paper handbook in Japan. Because many patients have not held their past vaccination records at their visit, clinician could not confirm the patient's vaccination status. We added the explanation about "unknown number of vaccinations" in result section. 

(Before)

The number of AR who received none, one-time, and unknown number of vaccinations were 11 (13.4%), 8 (9.8%), and 63 (76.8%), respectively.

(After)

The number of AR who received none, one-time, and unknown number of vaccinations were 11 (13.4%), 8 (9.8%), and 63 (76.8%), respectively. Unknown number of vaccinations means that clinician could not confirm the patient's vaccination status.

d. Page 19, line 223, you present genotype 1E, as the most common. Nevertheless, you do not describe genotyping in the methods section.

Thank you for your suggestion. However, we respectfully disagree with the reviewer. As we mentioned in Laboratory analysis, genotyping was conducted according to the pathogen detection manual of National Institute of Infectious Diseases in Japan. This manual is based on the standardization of the nomenclature for wild type rubella virus determined by WHO. Rubella virus E1 genotyping refer to the gene sequence of the window region (739 bp, positions 8731-9469) in the gene [18]. We already refer to this paper in the manuscript.

e. Please clarify the high-risk population group in the section: “study design and sampling”.

Thank you for your advice. high-risk population group defined as Japanese men who were born from 1962 to 1979. These men were supposed to have low immunity rubella (about 80%) because they were not eligible for the national regular rubella vaccination program. Although we described about high-risk population in the statistical analysis, we moved to study design and sampling section based on your advice. 

(Before: Study design and sampling)

Eligible subjects were those with suspected symptomatic rubella, aged ≥ 18 years who visited NCGM and were screened using rubella-specific IgM test with enzyme immune assay (EIA) kit. The following exclusion criteria were applied: (i) all patients aged < 18 years; (ii) clinically suspected rubella, which resulted in unconfirmed diagnosis due to weak or negative rubella-specific IgM.

(After: Study design and sampling)

Eligible subjects were those with suspected symptomatic rubella, aged ≥ 18 years who visited NCGM and were screened using rubella-specific IgM test with enzyme immune assay (EIA) kit. The following exclusion criteria were applied: (i) all patients aged < 18 years; (ii) clinically suspected rubella, which resulted in unconfirmed diagnosis due to weak or negative rubella-specific IgM. We defined Japanese men born from 1962 to 1979 as high-risk population because they were not eligible for the national regular rubella vaccination due to the national vaccination program in Japan. The antibody titer for this population was low (about 80%) compared to that of the other generation (over 90%) [14].

 (Before: Statistical analysis)

The sub-analysis was conducted among high-risk population (men born from 1962 to 1979) of AR. Because this population was not eligible for the national regular rubella vaccination due to the national vaccination program in Japan, the antibody titer for this high-risk population was low (about 80%) compared to that of the other population (over 90%) [8, 16].

(After: Statistical analysis)

The sub-analysis was conducted among high-risk population (Japanese men born from 1962 to 1979) of AR. 

3. Results/Tables

a. Table 1. Not easy to read. More readable if you divide it in 2 tables. 

Thank you for your advice. We divided Table1 in terms of the backgrounds and clinical characteristics as showed below. 

(Before)

Table 1 Univariate and multivariate analysis of clinical characteristics of adult rubella, n=221

Table 2 Multivariate analysis of the characteristics among high-risk population* of adult rubella, n=68

Table 3 Univariate analysis of the characteristics among adult rubella patients between none and one-time vaccination, n=19

(After)

Table 1 Univariate and multivariate analysis of backgrounds of adult rubella, n=221

Table 2 Univariate and multivariate analysis of clinical characteristics and laboratory findings of adult rubella, n=221

Table 3 Multivariate analysis of the characteristics among high-risk population* of adult rubella, n=68

Table 4 Univariate analysis of the characteristics among adult rubella patients between none and one-time vaccination, n=19

b. The same information is repeated in the manuscript, could be removed from the tables or from the text.

Thank you for your advice. Based on reviewer’s comment, we removed repeated sentence with no significance from the manuscript in result section.

(Before: Comparison of clinical characteristics between AR and ANR)

As shown in Table 1, the median (IQR) age of patients with AR and ANR was 31 (25–41) years and 34 (27–42) years, respectively. Most patients with AR (98.8%) and ANR (94.2%) were Japanese. No pregnant patients were observed. The number of AR who received none, one-time, and unknown number of vaccinations were 11 (13.4%), 8 (9.8%), and 63 (76.8%), respectively. The major symptom found in this study population was rash (100% [82/82] in AR and 87.8% [122/139] in ANR).

(After: Comparison of clinical characteristics between AR and ANR)

As shown in Table 1 and Table 2, the median (IQR) age of patients with AR and ANR was 31 (25–41) years and 34 (27–42) years, respectively. The number of AR who received none, one-time, and unknown number of vaccinations were 11 (13.4%), 8 (9.8%), and 63 (76.8%), respectively. The major symptom found in this study population was rash (100% [82/82] in AR and 87.8% [122/139] in ANR).

(Before: Comparison of clinical characteristics between AR and ANR )

At univariate analysis, AR compared to ANR, was significantly associated with male sex (78% vs. 56.1%, OR = 2.8; 95% CI = 1.5–5.2; P = 0.001) and pre-exposure to other rubella patients (OR = 4.2; 95% CI = 1.2–14.0; P = 0.016). Otherwise, they were less likely to have travel history (OR = 0.3; 95% CI = 0.1–0.8; P = 0.008).

(After: Comparison of clinical characteristics between AR and ANR)

At univariate analysis, AR compared to ANR, was significantly associated with male sex (78% vs. 56.1%, OR = 2.8; 95% CI = 1.5–5.2; P = 0.001) and pre-exposure to other rubella patients (OR = 4.2; 95% CI = 1.2–14.0; P = 0.016). 

c. Please add as a new table the comparative results, with statistical analysis, of the two periods of your study.

Thank you for your suggestion. However, we respectfully disagree with the reviewer. Number of cases and controls during two periods were already described in the result section (Line 197-198). We also conducted univariate and multivariate analysis including these two periods between adult rubella and adult non-rubella as shown in Table1, but we couldn’t statistical difference in multivariate analysis (OR=0.5, 95%CI=0.1-2.3, p value=0.351). Therefore, it seemed not so significant to add a new table describing the comparative results of the two period. We are also concerned an additional table makes the manuscript more complicating because we already had four tables in this article.

4. It is very interesting the main finding of conjunctivitis in AR. Is it possible to conduct a pathogenetic explanation? In conjunction with congenital rubella, where conjunctivitis is also a main clinical symptom.

Thank you for your advice. Rubella seems to spread in lymphatic tissue and produce follicular conjunctivitis as many other viral conjunctivitis. Pink book from CDC describes “Following respiratory transmission of rubella virus, replication of the virus is thought to occur in the nasopharynx and regional lymph nodes” and Mandell, Douglas & Bennett’s 9th says “Rubella produces a catarrhal or follicular reaction, or both, along with the typical disease findings.” So, we think that conjunctivitis in AR are thought to be from follicular reaction including follicular conjunctivitis along with catarrhal symptoms. We added with new refences [19, 20] in discussion section. 

(Before)

Limited clinical studies have evaluated for conjunctivitis among AR. The only past case series of rubella outbreak during 2012–2013 in Japan reported that conjunctivitis was observed in 77.8% of adult patients [10].

(After)

Limited clinical studies have evaluated for conjunctivitis among AR. The only past case series of rubella outbreak during 2012–2013 in Japan reported that conjunctivitis was observed in 77.8% of adult patients [10]. We think that rubella produces follicular reaction including follicular conjunctivitis along with catarrhal symptoms [19, 20].

19. Centers for Disease Control and Prevention, USA. CDC; The Pink Book, Chapter 20; Rubella. [cited 2020 March 14]. Available from: https://www.cdc.gov/vaccines/pubs/pinkbook/rubella.html

20. Kumar DM, Barnes SD, Pavan-Langston D, Azar DT. Microbial Conjunctivitis. In Bennett JE, Dolin R, Blaser MJ, editors. Mandell, Douglas and Bennett's Principles and Practice of Infectious Diseases 9th ed; 2019. p. 1502.

5. Regarding your result of not known vaccinations, it could be added in the discussion section that a further investigation, retrospectively, of the immunization status of this subjects, would be very informative about the need for two dose vaccination scheme.

Thank you for your suggestion. In Japan, vaccination status was recorded in the mother and child health paper handbook. This handbook was personally kept, and if patients did not bring this handbook during the hospital visit, clinicians did not refer to the vaccination status. Patients with unknown vaccination, as mentioned above, were those who clinician could not confirm the patient's vaccination status at patient's visit. Because our study is retrospective research, we cannot get any more information about vaccination and immunization status. However, our study showed that there were no rubella patients who had two-time vaccination. Two-time vaccination is thought to be significant to prevent rubella. We emphasize this point in the discussion part.

(Before)

There were no differences in clinical characteristics between AR who received one-time vaccination and unvaccinated patients. No data reflected symptoms of rubella in both children and adults pertaining to the vaccination status. Rubella vaccine was considered as highly immunogenic [17, 18], but 8 (9.8%) AR patients had received one-time vaccination in our study. Otherwise, no two-time vaccination was observed in our study. To prevent rubella, not only one- but two-time vaccinations are needed. To stop the ongoing rubella outbreak, Japanese government started an evaluation of an antibody test and catch-up vaccination program for high-risk population since April 2019 [19].

(After)

There were no differences in clinical characteristics between AR who received one-time vaccination and unvaccinated patients. No data reflected symptoms of rubella in both children and adults pertaining to the vaccination status. Rubella vaccine was considered as highly immunogenic [21, 22], but 8 (9.8%) AR patients had received one-time vaccination in our study. Otherwise, no two-time vaccination was observed in our study. Although further information about vaccination and immunization status among patients with unknown vaccination were not collected due to retrospective research, our study showed that not only one- but two-time vaccinations were needed to prevent rubella. To stop the ongoing rubella outbreak, Japanese government started an evaluation of an antibody test and catch-up vaccination program for high-risk population since April 2019 [23].

6. A few spelling and grammatical suggestion

a. Line 33 add comma after kit

b. Line 81 change with to by

c. Line 127 add comma before and after “using an EIA kit”

d. Line 135 replace one with last

e. Line 138 remove defend as

f. Line 139 and nausea change to: nausea;

g. Line 140 change test and enzyme to tests and enzymes respectively

Thank you for your advice. We revised grammatical errors that you pointed out, as bellow.

a. Line 33 add comma after kit

(Before)

enzyme immune assay kit at a tertiary care hospital in Japan during two outbreaks

(After)

enzyme immune assay kit, at a tertiary care hospital in Japan during two outbreaks

b. Line 81 change with to by

(Before)

Although rubella in children is characterized with

(After)

Although rubella in children is characterized by

c. Line 127 add comma before and after “using an EIA kit”

(Before)

and negative titers of rubella-specific IgM test using an EIA kit were

(After)

and negative titers of rubella-specific IgM test, using an EIA kit, were

d. Line 135 replace one with last

(Before)

travel history within one month

(After)

travel history within last month

e. Line 138 remove defend as

(Before)

catarrhal symptoms (defined as cough,

(After)

catarrhal symptoms (cough,

f. Line 139 and nausea change to: nausea;

(Before)

and nausea and vomiting

(After)

nausea and vomiting

g. Line 140 change test and enzyme to tests and enzymes respectively

(Before)

laboratory test including complete blood cell counts with atypical lymphocyte, liver enzyme,

(After)

laboratory tests including complete blood cell counts with atypical lymphocyte, liver enzymes,

Reviewer #2

1. In this MS, the statistic is by comparing the AR with ANR; however, it is not well defined how these patients (AR+ANR) were selected. This information is very important and might create a bias for the statistic, authors should explain how these cohort were selected. 

Thank you for your advice. As well as response to Reviewer # 1's comment (2-a), first, we included these study patients (AR+ANR) with suspected symptomatic rubella patients based on clinical symptoms such as fever or rash or lymphadenopathy which are described in the Infectious Disease Surveillance System in Japan [13]. Second, we confirmed the rubella using specific IgM antibodies for rubella in serum and RT-PCR. According to your suggestion, we added the additional explanation about how these patients (AR+ANR) were selected with new reference [13] in method part.

(Before: Study design and sampling)

A retrospective observational study of all symptomatic patients suspected of having rubella was conducted during two outbreaks (January 2012–December 2013 and January 2018–March 2019) at NCGM, Japan.

(After: Study design and sampling)

A retrospective observational study of all symptomatic patients suspected of having rubella, based on clinical symptoms such as fever or rash or lymphadenopathy which are described in the Infectious Disease Surveillance System in Japan [13], was conducted during two outbreaks (January 2012–December 2013 and January 2018–March 2019) at NCGM, Japan.

(Before: Definition of adult rubella and adult non-rubella)

An AR patient was defined as an eligible subject who was confirmed as having rubella on account of the following criteria (based on rubella-specific IgM test using an EIA kit and reverse-transcription-polymerase chain reaction (RT-PCR) test): (ⅰ) IgM showing strong positive result with a single serum at first hospital visit; (ⅱ) IgM showing negative or weak result at first hospital visit, but changed to strong positive at follow-up visit; (ⅲ) RT-PCR of throat swab, carried out by the local health government, showing positive rubella. Strong, weak, and negative titers of rubella-specific IgM test using an EIA kit were ≥ 1.21, 0.8–1.2, and < 0.8, respectively [13]. Adult non-rubella (ANR) patient was defined as an eligible subject without the evidence of rubella infection.

(After: Definition of adult rubella and adult non-rubella) 

First, we included these study patients with suspected symptomatic rubella based on clinical symptoms such as fever or rash or lymphadenopathy which are described in the Infectious Disease Surveillance System in Japan [13]. Second, we confirmed the rubella using specific IgM antibodies for rubella in serum and reverse-transcription-polymerase chain reaction (RT-PCR) test. An AR patient was defined as an eligible subject who was confirmed as having rubella on account of the following criteria: (i) IgM showing strong positive result with a single serum at first hospital visit; (ii) IgM showing negative or weak result at first hospital visit, but changed to strong positive at follow-up visit; (iii) RT-PCR of throat swab, carried out by the local health government, showing positive rubella. Strong, weak, and negative titers of rubella-specific IgM test using an EIA kit were ≥ 1.21, 0.8–1.2, and < 0.8, respectively [15]. ANR patient was defined as an eligible subject without the evidence of rubella infection.

13. National Institute of Infectious Disease in Japan. Infectious Disease Surveillance System in Japan. [cited 2020 March 14]. Available from: https://www.niid.go.jp/niid/images/epi/nesid/nesid_en.pdf

2. Follow the same idea, I am not sure if it is sufficient only compared the AR and ANR can draw a conclusion. As it is well know, Lymphadenopathy is a key phenotype of Rubella infection, indeed, this phenotype is common in both AR and ANR group, this suggested that lymphadenopathy is being considered and therefore these patients were further received the Rubella screening. This will also cause the bias; if authors only compared AR and ANR groups, authors should emphasis this point through the MS that this is pre-selected cohort analysis.

Thank you for your advice. As mentioned above, the inclusion criteria of this research patients (AR+ANR) were suspected symptomatic rubella patients based on clinical symptoms such as fever or rash or lymphadenopathy which are described in the Infectious Disease Surveillance System in Japan, and they were evaluated rubella infection using laboratory examination, including RT-PCR and rubella IgM antibody test. Therefore, rubella-suspected symptoms such as lymphadenopathy were likely to be pre-selected bias. We added the limitation about pre-selected bias in discussion part.

(Before)

Finally, we did not evaluate for difference in conjunctivitis between AR and measles due to the low number of measles cases (n=5). Conjunctivitis has been reported as one of the major clinical characteristics of measles [25]; however, only 2 of 5 measles cases showed conjunctivitis in our study. There is the possibility that the clinical symptoms were underestimated due to the retrospective nature of the study. To evaluate the clinical characteristics between AR and measles, further study is needed.

(After)

Fourth, we did not evaluate for difference in conjunctivitis between AR and measles due to the low number of measles cases (n=5). Conjunctivitis has been reported as one of the major clinical characteristics of measles [28]; however, only 2 of 5 measles cases showed conjunctivitis in our study. There is the possibility that the clinical symptoms were underestimated due to the retrospective nature of the study. To evaluate the clinical characteristics between AR and measles, further study is needed. Finally, the inclusion criteria of these study patients (AR and ANR) were suspected symptomatic rubella patients based on clinical symptoms such as fever or rash or lymphadenopathy which are described in the Infectious Disease Surveillance System in Japan, and they were evaluated rubella infection using laboratory examination, including RT-PCR and rubella IgM antibody test. Therefore, rubella-suspected symptoms such as lymphadenopathy were likely to be pre-selected bias in this research. However, our new finding "conjunctivitis, the key clinical characteristic of adult rubella" is thought to be important to distinguish diseases within suspected symptomatic rubella patients at clinical practice.

3. I am very confused about the data presented in Tables. For example, in table I, Trunk 81 (98.8%); extremity 82 (100%); but conjuctivitis 68 (94.4%) and peri-auricular 30 (57.7%). Could authors explain this point? this is extremely important, since this will large affect the conclusion. I suggest that 94.4% is 68 cases have conjuctivitis and 4 cases did not. it means that only 72 in total instead of 82. I don't know why 10 cases were exclude?

Thank you for your suggestion. Due to retrospective study, if data were not listed in the electronic medical record, we treated as missing values, and were removed from the whole number (both numerator and denominator). So, the values in the table were not consistent. To emphasize this point, we added the further explanation in methods section.

(Before)

All eligible subjects who were screened for rubella infection were identified through the hospital laboratory database. The parameters retrieved from patients’ records included the following; (i) demographics including age, sex, nationality, pre-exposure to other rubella patients, travel history within one month, pregnancy, number of days from onset to hospital visit; (ii) vaccination status; (iii) rubella-specific IgM serology at first visit; (iv) clinical symptoms including maximum temperature (fever) from onset to the visit; presence and location of rash and lymphadenopathy; conjunctivitis; catarrhal symptoms (defined as cough, pharyngitis, and rhinitis); arthralgia; headache; diarrhea; and nausea and vomiting; (v) laboratory test including complete blood cell counts with atypical lymphocyte, liver enzyme, lactate dehydrogenase (LDH), C-reactive protein (CRP); and (iv) virus subtype.

(After)

All eligible subjects who were screened for rubella infection were identified through the hospital laboratory database. The parameters retrieved from patients’ records included the following; (i) demographics including age, sex, nationality, pre-exposure to other rubella patients, travel history within last month, pregnancy, number of days from onset to hospital visit; (ii) vaccination status; (iii) rubella-specific IgM serology at first visit; (iv) clinical symptoms including maximum temperature (fever) from onset to the visit; presence and location of rash and lymphadenopathy; conjunctivitis; catarrhal symptoms (cough, pharyngitis, and rhinitis); arthralgia; headache; diarrhea; nausea and vomiting; (v) laboratory tests including complete blood cell counts with atypical lymphocyte, liver enzymes, lactate dehydrogenase (LDH), C-reactive protein (CRP); and (iv) virus subtype. If data were not listed in the electronic medical record, we treated these as missing values, and were removed from the whole number (both numerator and denominator), due to retrospective study.

---

## [Decision Letter · Decision Letter 1]

6 Apr 2020

Conjunctivitis, the key clinical characteristic of adult rubella in Japan during two large outbreaks, 2012-2013 and 2018-2019

PONE-D-19-34587R1

Dear Dr. Ishikane,

We are pleased to inform you that your manuscript has been judged scientifically suitable for publication and will be formally accepted for publication once it complies with all outstanding technical requirements.

With kind regards,

Giovanna Barba-Spaeth, Ph.D.

Academic Editor

PLOS ONE

Additional Editor Comments (optional):

Reviewers' comments:

Reviewer's Responses to Questions

**Comments to the Author**

1. If the authors have adequately addressed your comments raised in a previous round of review and you feel that this manuscript is now acceptable for publication, you may indicate that here to bypass the “Comments to the Author” section, enter your conflict of interest statement in the “Confidential to Editor” section, and submit your "Accept" recommendation.

Reviewer #2: All comments have been addressed

2. Is the manuscript technically sound, and do the data support the conclusions?

Reviewer #2: Yes

3. Has the statistical analysis been performed appropriately and rigorously? 

Reviewer #2: Yes

4. Have the authors made all data underlying the findings in their manuscript fully available?

Reviewer #2: Yes

5. Is the manuscript presented in an intelligible fashion and written in standard English?

Reviewer #2: Yes

6. Review Comments to the Author

Reviewer #2: (No Response)

7. PLOS authors have the option to publish the peer review history of their article (what does this mean?). If published, this will include your full peer review and any attached files.

Reviewer #2: No

---

## [Editor Report · Acceptance letter]

10 Apr 2020

PONE-D-19-34587R1 

Conjunctivitis, the key clinical characteristic of adult rubella in Japan during two large outbreaks, 2012–2013 and 2018–2019 

Dear Dr. Ishikane:

I am pleased to inform you that your manuscript has been deemed suitable for publication in PLOS ONE. Congratulations! Your manuscript is now with our production department. 

With kind regards,

on behalf of

Dr. Giovanna Barba-Spaeth 

Academic Editor

PLOS ONE